# The Influence of the Burnishing Process on the Change in Surface Hardness, Selected Surface Roughness Parameters and the Material Ratio of the Welded Joint of Aluminum Tubes

**DOI:** 10.3390/ma17010043

**Published:** 2023-12-21

**Authors:** Wojciech Labuda, Agata Wieczorska, Adam Charchalis

**Affiliations:** Faculty of Marine Engineering, Gdynia Maritime University, Morska St. 81-87, 81-225 Gdynia, Poland; a.wieczorska@wm.umg.edu.pl (A.W.); a.charchalis@wm.umg.edu.pl (A.C.)

**Keywords:** EN AW6060 alloy, aluminum welding, burnishing, surface roughness, material ratio, surface layer hardening

## Abstract

This paper presents the effect of burnishing on the surface hardness, selected surface roughness parameters and material ratio of tubes made of an EN AW-6060 aluminum alloy after welding. The prepared specimens were subjected to a 141-TIG welding process, after which the surfaces to be burnished were given a finishing turning treatment with DURACARB’s CCGT09T302-DL cutting insert to remove the weld face. After the turning process, the surface finish treatment was carried out by rolling burnishing, for which Yamato’s SRMD burnishing tool was used. The surface hardness, selected surface roughness parameters and material ratio were then measured. An analysis of the results showed an increase in hardness in the surface layer, as well as an improvement in the analyzed surface roughness parameters and the material ratio of the native material and the weld.

## 1. Introduction

Aluminum alloys are ideal materials to replace heavier materials such as steel or copper due to their properties, i.e., function applied to the case, high strength and stiffness to the weight, good formability, strength and applicability [1,2,3]. Civil engineering structures and tools made of aluminum alloys are being increasingly used [4]. The AW-6060 T66 aluminum stopper belongs to the aluminum–magnesium–silicon family (6000 or 6xxx series). Alloys belonging to the 6000 series include the amount of minerals and silicon as well as additives such as Mn, Cu, Cr and Zn. They contain approximately 1% clarification and silicon, which is used as clarification with the 4000 series alloys, which are silicon-infused (up to 13% Si), and the 5000 series alloys, which are magnesium-infused (up to 5% Mg). The addition of silicon increases the efficiency and improves the performance. Aluminum in the publication with magnesium and silicon forms a pseudo-binary system Al-Mg_2_Si. The Mg_2_Si intermetallic phase is responsible for the precipitation hardening of these alloys. Depending on the alloy components, the 6000 family alloys may also include AlMg_2_Mn, Al_3_Mg_2_ and CuAl_2_ phases. These alloys are characterized by durability class B according to the standard [4,5]. Thanks to this principle, the application and protection required are possible, i.e., for a 3 mm thick material as no protection against occurring is usually required. The EN AW-6060 alloy is a widely available alloy that can be used in thermal equipment and often primary structural alloys for welded and non-welded applications. They come in various forms such as extruded tubes, profiles, rods and drawn tubes. Stop ten elements is an element for which the durability of the structure, surface finish, as well as the possibility of performing functional functions and thin-walled profiles are important. It is an element suitable for anodizing and basic treatment. This alloy is easily weldable in the tungsten inert gas (TIG) and metal inert gas (MIG) welding processes, but traction in the heat-affected zone (HAZ) [4,5]. The performance resistance of welded joints made of steel and aluminum is included in the detailed composition. This effect is a notch effect available through the weld bead, a change in the microstructure (metallic notch) and access to residual resources originating from the weld root. As a result of these recent results, many post-weld impact methods have been developed and validated to achieve power design, exemplified by a high-frequency mechanical impact [6]. Rodríguez A., Calleja A. et al. [7] tested a 2050 aluminum alloy after the friction stir welding (FSW) process and then performed the ball burnishing process. Results were obtained regarding surface roughness values that are available on FSW and burnishing surfaces. The surface hardness can be increased up to 60% for available aluminum alloys. In the case of aluminum without heat application, the ball burnishing effect was additional in the joints, where the material was softer and, consequently, more hardenable via plastic deformation. Schubnell J., Farajian M. [8] tested the EN AW 5083 aluminum alloy after the diamond ball burnishing process of the joints via laboratory tests using the Ecoroll hydrostatic tool. They had the advantage of roughness and additional hardness features at the weld root. Baisukhan A., Nakkiew W. [9] carried out the shot-rolling processing of a 7075-T6 aluminum alloy, and the emitter of the function was a sintered carbide roller with a conical and rounded contact. The authors investigated the distribution of residual stresses induced by deep rolling, the surface roughness and hardness distribution. It was found that the initial surface preparation has an additional influence on the surface roughness than the rolling parameters, and an increase in the hardness was achieved. The claim also belongs to a very small feed because it is caused by strong work hardening with limited access to the surface and subsurface residual stresses that are limited by the yield strength.

Burnishing [10] is a treatment aimed mainly at improving the surface finish and/or improving the surface hardness [11,12,13], where the plastic deformation is limited to the scale of the surface roughness. This treatment is not primarily intended to introduce residual stresses. On the other hand, deep rolling [14] (sometimes also referred to as “deep cold rolling” [15] or “deep ball burnishing” using a spherical indenter [16]) is a treatment essentially designed to introduce surface and subsurface highly compressive residual stresses. Improvements in the surface roughness are still obtained even in deep rolling, along with high cold working [15,17,18]. The latter surface effect can be beneficial as it relates to the improvement in the hardness, but also harmful because it causes the brittleness of the material [19,20]; these two controversial factors can play a relative role depending on the applied load and the specific material. Low plasticity burnishing, developed and patented by Lambda Technologies, also referred to as “roller burnishing” e.g., by Ecoroll Company and Klocke and Liermann [21] or “ball burnishing” by López et al. [22], is intended to reduce both the surface roughness and high and deep compressive residual stresses. The main difference concerning deep rolling is that high residual stresses are obtained with reduced work hardening [23,24,25] mainly due to the large size of the spherical indenter. This type of burnishing is usually performed on ordinary machine tools, with a hydrostatic bearing system for the ball roller, which requires a pressure unit. Even complex geometries can be processed using this technology. The authors Kluz R., Bucior M., et al. [26] investigated the sliding diamond burnishing process at variable feed rates of butt joints made by the FSW method of EN AW-2024 aluminum alloys. They obtained a significant reduction in the roughness and an increase in the microhardness was observed. The burnishing treatment generated compressive residual stresses, increasing the fatigue strength of the joints. In contrast, Teimouri R., Grabowski M., Bogucki R., et al. [27] developed a physics-based analytical model to conclude that the hardening mechanism contributes to the improvement in the surface properties during the pressing process. The results show good agreement between the measured and predicted hardness values. Furthermore, the depth of pressing was identified as the most influential parameter affecting the hardness and hardening depth. The results of the conducted research by Felhȍ C., Varga G. [28] indicate that the diamond burnishing process improved the residual stress properties of EN1.4301 austenitic stainless steel by creating a relatively high compressive stress, whose magnitude was between 629 and 1138 MPa depending on the applied force. However, the stress distribution is not uniform; it is mostly concentrated under the roughness peaks. Jerez-Mesa et al. [29] investigated the ultrasonic vibration-assisted ball burnishing process and how to develop a vibration-free version of it, as well as the consequences for the topology and subsurface microstructure of the concrete workpiece.

Many manufacturers of burnishing tools and numerous research centers around the world conduct research related to the process of burnishing parts and components of machines and equipment that are made of various construction materials. Researchers are focusing on the introduction of new design solutions for both burnishing tools and methods of carrying out the surface treatment process. An analysis of the literature has shown that it is appropriate to address the issue of the burnishing of welded joints of aluminum tubes made of an EN AW-6060 alloy. This study aimed to determine the effect of burnishing, carried out on a conventional lathe, on the degree of surface hardness and the reduction rates of selected surface roughness parameters and the material ratio.

## 2. Materials and Methods

EN AW6060 aluminum alloy tubes, which were given a welding process followed by a finishing turning process with surface treatment, were used to carry out this study. Aluminum tubes with a diameter of 60 mm and a wall thickness of 5 mm were subjected to the 141–TIG (argon 4.5 gas) welding process. Three specimens were prepared to carry out the tests. The welding process parameters are presented in Table 1, while Table 2 shows the chemical composition and mechanical properties of the filler metal.

The conventional TUC 50 × 1000 lathe (Wafum, Wroclaw, Poland) was used for the tests (Figure 1). To ensure proper mounting of the sample, the rotary center was used as the lathe drive dog (Figure 2). The preliminary and finishing turning processes were carried out using a replaceable cutting insert CCGT09T302-DL from DURACARB (DURACARB, Dalian, China).

Initial turning was carried out to remove the weld face, while the next pass was a finishing pass, which was carried out with the following cutting parameters: cutting speed V_c_ = 211 m/min, feed f = 0.08 mm/rev and cutting depth a_p_ = 0.5 mm. The turning process was carried out using a cooling and lubricating liquid. The view of the mounted workpiece and the lathe cutter is shown in Figure 3a. The burnishing process was carried out using a Yamato SRMD burnishing tool (Yamato Ltd., Bornova, Turkey). The view of the mounted sample in the centers and the burnishing tool in the tool post is shown in Figure 3b. The burnishing parameters were selected based on technological documentation and selected tests carried out for 304L stainless steel [32,33] and for the tested EN AW-6060 alloy [34]. The parameters used in the surface forming process are burnishing force F_n_ = 1.0 kN, feed f_n_ = 0.08 mm/rev and burnishing speed V_n_ = 42 m/min. The burnishing speed and feed values were selected considering the lathe’s setting capabilities. Machine oil was used during the burnishing process.

Hardness measurements were made using the Vickers HV10 method and were carried out on the Qness 250M (ATM Qness GmbH, Mammelzen, Germany) universal hardness tester (Figure 4a). Figure 4b shows a view of one measurement. To determine the effect of the burnishing treatment on the hardening of the surface layer, the relative surface hardening index S_u_ [%] was determined (1).
(1)$$S_{u}=\frac{{\mathrm{HV}}_{2}-{\mathrm{HV}}_{1}}{{\mathrm{HV}}_{1}}\times 100\%,$$
where
S_u_—index of relative surface strengthening;HV_1_—material hardness before burnishing;HV_2_—material hardness after burnishing.

**Figure 4 materials-17-00043-f004:**
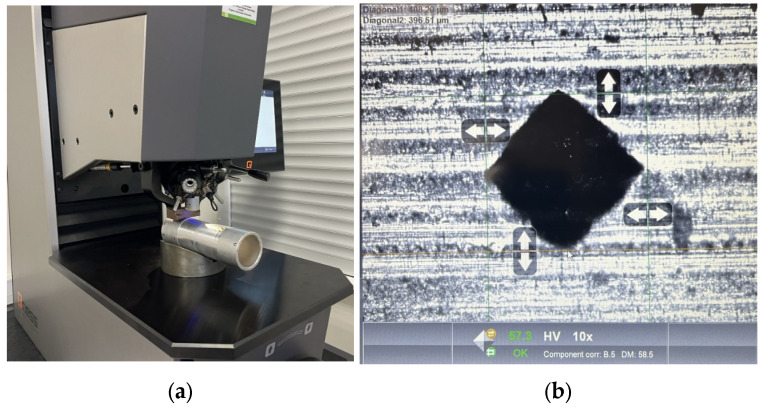
Hardness measurement. (**a**) Qness 250M hardness tester, (**b**) sample measurement results.

The measurement of selected surface roughness parameters and the material ratio was performed using a Hommel-Etamic W20 profilometer (JENOPTIK Industrial Metrology Germany GmbH, Tönisvorst, Germany) (Figure 5). The measurement of the analyzed parameters was carried out for the elementary section lr = 0.8 mm and the measurement section ln = 4.0 mm, with the mapping section set to 4.8 mm. During the tests, a measurement speed of V_t_ = 0.5 mm/s was used. The following parameters were used to determine the effect of the burnishing treatment on the change in the height of surface irregularities:
R_a_ [μm]—arithmetic mean deviation;R_q_ [μm]—root mean square slope;R_t_ [μm]—total height of profile;R_z_ [μm]—maximum height of profile;R_k_ [μm]—core roughness depth;R_pk_ [μm]—reduced peak height;R_pv_ [μm]—reduced valley depth.

**Figure 5 materials-17-00043-f005:**
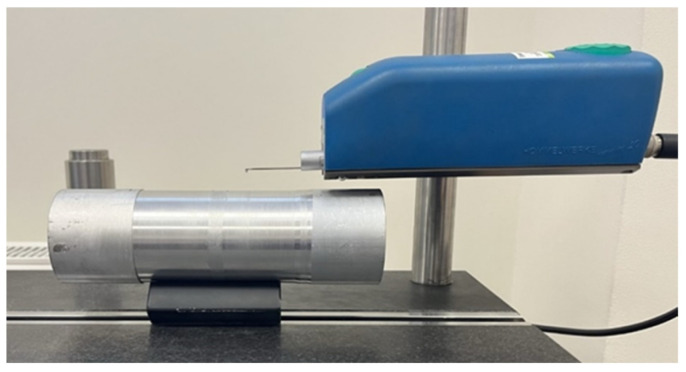
Surface roughness measurement using the W20 profilometer.

To analyze the impact of the burnishing treatment on the obtained measurement results, the indicators of surface roughness reduction and material ratio were used. The example formula (2) determines the reduction index for the roughness parameter R_a_, while the same relationship was used for the remaining parameters.
(2)$$KR_{a}=\frac{R_{a2}}{R_{a1}}$$
where

KR_a_—surface roughness reduction index for the R_a_ parameter;R_a2_—surface roughness parameter after burnishing [μm];R_a1_—surface roughness parameter after turning [μm].

## 3. Results and Discussion

Measurements of the parameters of the material ratio, roughness and surface hardness of the tested welded joint were performed at three measurement locations on the tube circumference, at intervals of 120°. The measurements were made after the turning process and burnishing. Figure 6 shows the sample measurement areas. The measurement tables include markings for individual zones according to the following areas:

I—native material before the weld;II—weld;III—native material behind the joint.

**Figure 6 materials-17-00043-f006:**
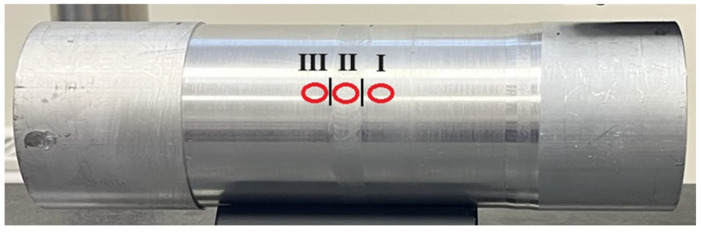
View of the measuring point of the tested sample.

Figure 7 shows a graph of the influence of the burnishing treatment on the change in the value of the roughness parameter R_a_. Figure 8 and Figure 9 present the surface roughness profile and the material ratio curve for the measurement made in the weld.

Figure 7, Figure 8 and Figure 9 confirm the beneficial effect of the burnishing treatment on the quality of the machined surface. Table 3, Table 4, Table 5 and Table 6 present the results of the basic statistical analysis for the analyzed roughness parameters R_a_, R_q_, R_z_ and R_t_. The welded joint of the tubes made of the EN AW6060 alloy after finishing turning was characterized by an average R_a_ parameter value of 1.01 and 1.24 μm in the native material, while a value of 2.45 μm was obtained in the weld. The applied surface plastic treatment allowed the R_a_ parameter to be reduced below 1 μm, which allowed for obtaining high KR_a_ roughness reduction index values of 15.1 and 21.9 in the native material and 28.3 in the weld. The roughness reduction index for the R_q_ parameter also shows significant values of the KR_q_ index, which are 12.9 and 19.8 for the native material, and 23.6 for the weld. The height parameters defined in the elementary and measurement sections are also characterized by a significant improvement in values after the burnishing treatment, as evidenced by the results presented in Table 5 and Table 6.

Figure 8 and Figure 9 show the material ratio curves after the turning and burnishing processes. The comparison of the course of these curves proves the beneficial effect of the burnishing process on reducing the peaks of the surface irregularities and increasing the load-bearing capacity of the surface. The burnishing roller reduces the peaks resulting from turning, as indicated by the obtained average values of the R_pk_ parameter. As the value of the R_pk_ parameter decreased, the values of the R_k_ and R_vk_ parameters also changed. The plastically deformed peaks filled the deep cavities, causing a multiple reduction in the R_vk_ parameter. Small values of the analyzed parameters should be a measure of high resistance to corrosion, abrasion and contact fatigue. The results of the basic statistical analysis for the R_k_, R_pk_ and R_vk_ parameters are presented in Table 7, Table 8 and Table 9.

The burnishing process increased the hardness value of the surface of the welded joint. Figure 10 shows the average results of the surface hardness measurement after finishing turning and surface plastic processing. The burnishing treatment of the surface of the welded joint of an aluminum alloy allowed us to obtain a relative surface strengthening index of over 30% in the native material before the weld and in the weld area. However, in the native material behind the weld, the Su parameter obtained was equal to 45.7%. The surface treatment performed after finishing turning is characterized by a higher surface hardness value as a result of the work hardening of the surface layer. The results of the basic statistical analysis for the surface hardness are presented in Table 10.

The basic features of metals and their alloys include surface hardness and microhardness, which are directly related to the properties of the surface layer, such as resistance to frictional wear, fatigue strength, residual stresses and plasticity. The surface hardness depends on the type of material and the strengthening, thermal or thermo-chemical treatment carried out. The change in the hardness and microhardness in the burnishing process is influenced by the burnishing tool, which, under the action of force, causes local elastic and plastic deformations in the contact zone with the workpiece. The phenomenon of strengthening mainly occurs in the surface layer. The grains are fragmented, flattened and elongated in the direction of the greatest deformations, creating a crushed texture. The ability of a material to be strain-hardened depends largely on its structure and the value of the burnishing parameters. The burnishing process is limited due to the permissible degree of plastic deformation and exceeding it results in the technological cracking and flaking of the processed surface.

## 4. Conclusions

The research results presented in this article confirm the main purposes of using burnishing as a finishing treatment. The surface plastic processing carried out increased the quality of the surface while strengthening it. The process was carried out on a conventional lathe with burnishing parameters selected based on the analysis of technological documentation and our scientific research. To verify the correct mounting of the workpiece and the burnishing process, tests were performed on three samples. The test results obtained for individual joints of welded aluminum tubes confirmed the repeatability of the finishing turning and burnishing processes. The native material and the weld of the welded joint were characterized by a significant decrease in the values of the analyzed surface roughness parameters and material ratio while strengthening the surface layer. The presented test results demonstrate that the EN AW-6060 aluminum alloy can be burnished after welding. Soft materials pose problems when implementing this problem; therefore, further research is necessary to optimize the burnishing parameters and determine the repeatability of processing for variable conditions during the turning and burnishing process. Then, it will be necessary to determine the potential and operational properties of the surface layer.

## Figures and Tables

**Figure 1 materials-17-00043-f001:**
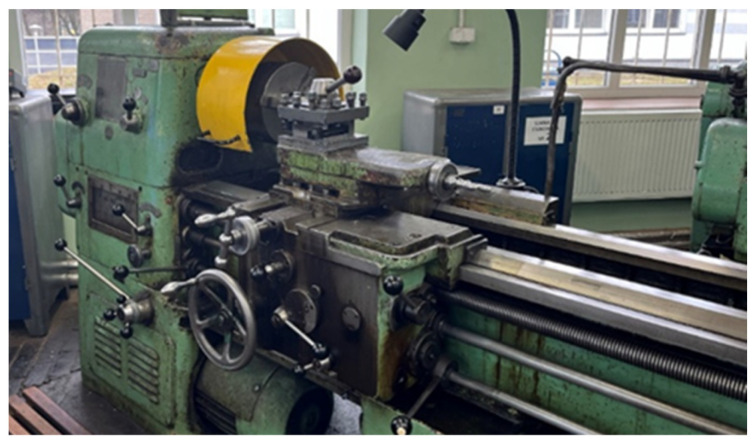
TUC 50 × 1000 lathe used in the tests.

**Figure 2 materials-17-00043-f002:**
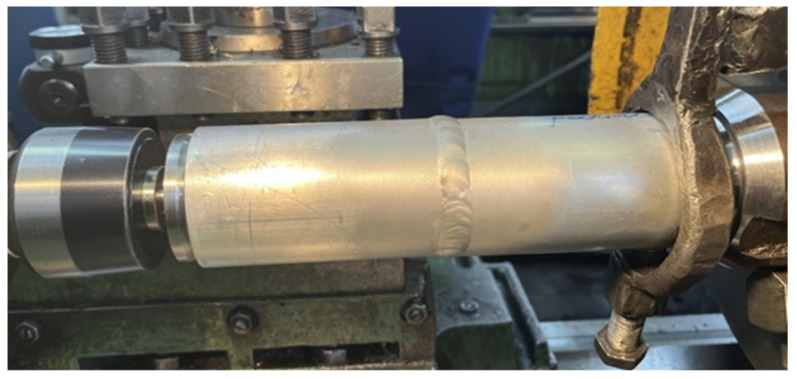
View of the welded sample mounting.

**Figure 3 materials-17-00043-f003:**
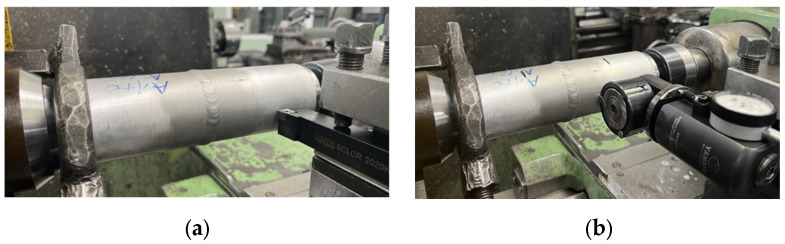
Fixing the tools used in the study. (**a**) The cutting tool used in the finishing turning process, (**b**) the burnishing tool used in the surface forming process.

**Figure 7 materials-17-00043-f007:**
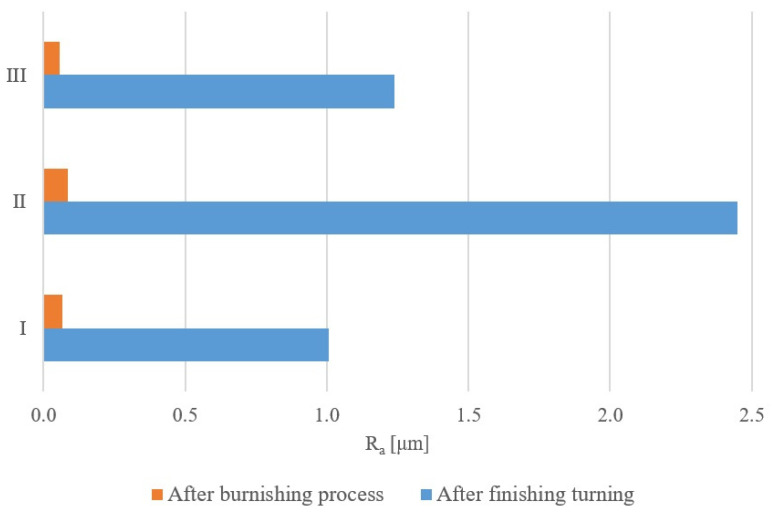
Change in the value of the Ra parameter after the burnishing process.

**Figure 8 materials-17-00043-f008:**
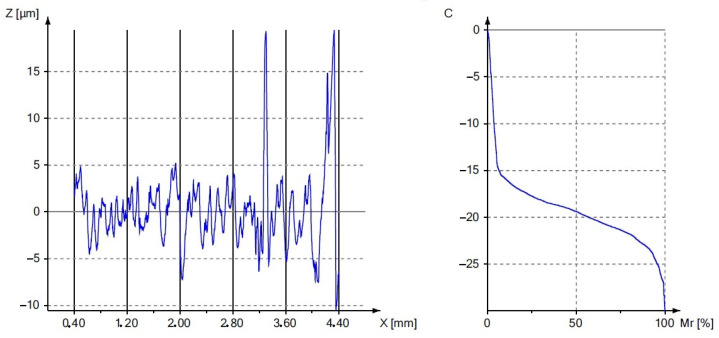
An example profile and material ratio curve after finishing turning.

**Figure 9 materials-17-00043-f009:**
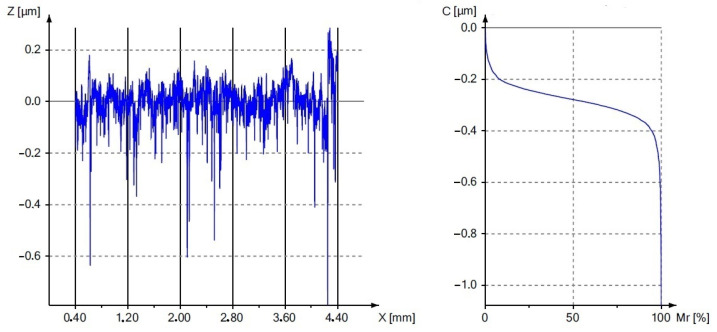
An example profile and material ratio curve after the burnishing process.

**Figure 10 materials-17-00043-f010:**
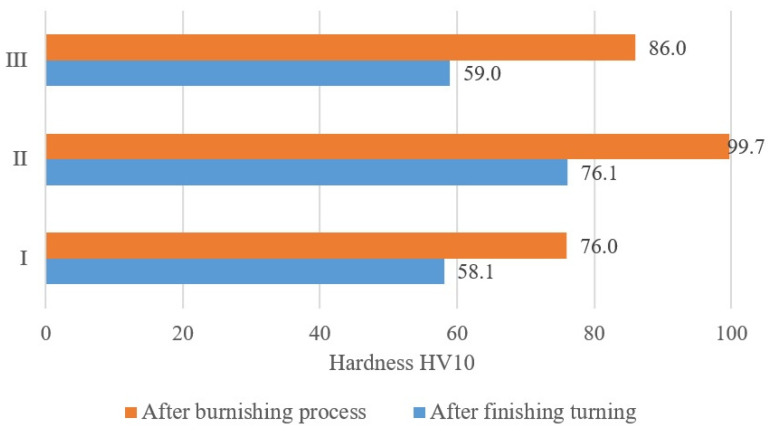
Change in the values of the hardness parameter after the burnishing process.

**Table 1 materials-17-00043-t001:** Welding process parameters.

Welding Process (by EN ISO 4063:2011 [30])	141 TIG
Joint type	BW (butt weld)
Welding position (by EN ISO 6947 [31])	PA (downhand position)
Base material	6060 AlMgSi_0.5_
Dimension of testing plates [mm]	L = 200 mm ø60
Filler material	ø3.2 mm OK Tigrod 5656
Arc Voltage U [V]	-
Welding Current I [A]	170- I bead penetration 120- II bead face of weld
Gas flow rate [L/min]	9

PA—downhand position—this is the most comfortable position for making any weld (whether using MMA welders, MIG/MAG semi-automatic welders or TIG equipment). The sub-lower position involves applying the electrode from above, preferably perpendicular to the welding surface. This position can be used on flat surfaces laid parallel to the ground, making the process very efficient. It is also a natural body position for the welder.

**Table 2 materials-17-00043-t002:** Chemical composition and mechanical properties of the filler metal.

**Supplementary** **Material**	**Al** **[%]**	**Si** **[%]**	**Mg** **[%]**	**Fe** **[%]**	**Mn** **[%]**	**Mechanical Properties**			
						**UTS** **[MPa]**	**YS** **[MPa]**	**EL** **[%]**	**AlMg_5_** **Ø 3.2 mm**
OK Tigrod 5356	95	<0.25	5.0	<0.40	<0.20	≥240	≥110	≥17	

**Table 3 materials-17-00043-t003:** Results of basic statistical analysis of R_a_ parameter and KR_a_ index.

Measurement Area	Basic Statistical Analysis	R_a_ [μm]		KR_a_ [-]
		After Turning	After Burnishing	
I	average value	1.01	0.07	15.1
	standard deviation	0.16	0.02	
	standard error	0.05	0.01	
	min	0.80	0.05	
	max	1.34	0.09	
II	average value	2.45	0.09	28.3
	standard deviation	0.31	0.04	
	standard error	0.10	0.01	
	min	1.98	11.99	
	max	2.87	17.04	
III	average value	1.24	0.06	21.9
	standard deviation	0.08	0.01	
	standard error	0.03	0.00	
	min	1.16	0.04	
	max	1.40	0.07	

**Table 4 materials-17-00043-t004:** Results of basic statistical analysis of R_q_ parameter and KR_q_ index.

Measurement Area	Basic Statistical Analysis	R_q_ [μm]		KR_q_ [-]
		After Turning	After Burnishing	
I	average value	1.25	0.10	12.9
	standard deviation	0.20	0.03	
	standard error	0.07	0.01	
	min	0.99	0.06	
	max	1.65	0.15	
II	average value	3.01	0.13	23.6
	standard deviation	0.41	0.06	
	standard error	0.14	0.02	
	min	2.25	0.07	
	max	3.63	0.27	
III	average value	1.50	0.08	19.8
	standard deviation	0.10	0.02	
	standard error	0.03	0.01	
	min	1.38	0.06	
	max	1.69	0.10	

**Table 5 materials-17-00043-t005:** Results of basic statistical analysis of R_t_ parameter and KR_t_ index.

Measurement Area	Basic Statistical Analysis	R_t_ [μm]		KR_t_ [-]
		After Turning	After Burnishing	
I	average value	7.90	1.39	5.7
	standard deviation	1.52	0.64	
	standard error	0.51	0.21	
	min	6.53	0.67	
	max	11.37	2.39	
II	average value	18.61	1.84	10.1
	standard deviation	5.67	1.17	
	standard error	1.89	0.39	
	min	14.57	0.79	
	max	29.97	4.59	
III	average value	8.17	0.92	8.9
	standard deviation	0.85	0.38	
	standard error	0.28	0.13	
	min	7.18	0.51	
	max	9.27	1.69	

**Table 6 materials-17-00043-t006:** Results of basic statistical analysis of R_z_ parameter and KR_z_ index.

Measurement Area	Basic Statistical Analysis	R_z_[μm]		KR_z_ [-]
		After Turning	After Burnishing	
I	average value	6.15	0.80	7.7
	standard deviation	0.87	0.29	
	standard error	0.29	0.10	
	min	4.94	0.48	
	max	7.78	1.26	
II	average value	13.23	1.00	13.2
	standard deviation	1.58	0.39	
	standard error	0.53	0.13	
	min	11.99	0.52	
	max	17.04	1.89	
III	average value	6.87	0.56	12.2
	standard deviation	0.50	0.24	
	standard error	0.17	0.08	
	min	6.33	0.18	
	max	7.88	0.99	

**Table 7 materials-17-00043-t007:** Results of basic statistical analysis of R_k_ parameter and KR_k_ index.

Measurement Area	Basic Statistical Analysis	R_k_ [μm]		KR_k_ [-]
		After Turning	After Burnishing	
I	average value	3.19	0.20	16.1
	standard deviation	0.53	0.03	
	standard error	0.18	0.01	
	min	2.61	0.16	
	max	4.19	0.25	
II	average value	7.69	0.25	30.5
	standard deviation	1.69	0.13	
	standard error	0.56	0.04	
	min	5.33	0.14	
	max	10.48	0.49	
III	average value	4.34	0.17	25.2
	standard deviation	0.26	0.03	
	standard error	0.09	0.01	
	min	4.09	0.13	
	max	4.88	0.22	

**Table 8 materials-17-00043-t008:** Results of basic statistical analysis of R_pk_ parameter and KR_pk_ index.

Measurement Area	Basic Statistical Analysis	R_pk_ [μm]		KR_pk_ [-]
		After Turning	After Burnishing	
I	average value	1.09	0.07	14.8
	standard deviation	0.27	0.02	
	standard error	0.09	0.01	
	min	0.78	0.05	
	max	1.64	0.11	
II	average value	3.04	0.10	30.0
	standard deviation	2.03	0.07	
	standard error	0.68	0.02	
	min	1.59	0.06	
	max	8.25	0.24	
III	average value	0.94	0.06	16.2
	standard deviation	0.17	0.01	
	standard error	0.06	0.00	
	min	0.71	0.04	
	max	1.22	0.07	

**Table 9 materials-17-00043-t009:** Results of basic statistical analysis of R_vk_ parameter and KR_vk_ index.

Measurement Area	Basic Statistical Analysis	R_vk_ [μm]		KR_vk_ [-]
		After Turning	After Burnishing	
I	average value	1.53	0.17	9.2
	standard deviation	0.60	0.08	
	standard error	0.20	0.03	
	min	0.86	0.08	
	max	2.89	0.30	
II	average value	3.40	0.25	13.5
	standard deviation	0.65	0.13	
	standard error	0.22	0.04	
	min	2.65	0.11	
	max	4.46	0.55	
III	average value	1.29	0.12	10.4
	standard deviation	0.18	0.05	
	standard error	0.06	0.02	
	min	1.06	0.08	
	max	1.57	0.20	

**Table 10 materials-17-00043-t010:** The results of the basic statistical analysis for the surface hardness.

Measurement Area	Basic Statistical Analysis	HV10		S_u_ [%]
		After Turning	After Burnishing	
I	average value	58.1	76.0	30.8
	standard deviation	3.9	2.9	
	standard error	1.3	1.0	
	min	50.7	72.5	
	max	63.9	80.6	
II	average value	76.1	99.7	31.0
	standard deviation	8.7	3.2	
	standard error	2.9	1.1	
	min	63.4	95.6	
	max	88.4	104.0	
III	average value	59.0	86.0	45.7
	standard deviation	3.2	4.2	
	standard error	1.1	1.4	
	min	53.2	80.1	
	max	61.7	92.0	

## Data Availability

Data are contained within the article.

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
