# Peer review of "The Influence of the Burnishing Process on the Change in Surface Hardness, Selected Surface Roughness Parameters and the Material Ratio of the Welded Joint of Aluminum Tubes"

_materials, 2023, doi:10.3390/ma17010043_

Round 1
Reviewer 1 Report
Comments and Suggestions for Authors
My suggestions and comments regarding your article are on file.

Author Response
Response to Reviewer Comments
Dear Reviewer,
Thank you very much for taking the time to review this manuscript. Thank You for all the comments that will certainly affect the quality of our article, the corrections have been made in the article.
The publication with the comments taken into account has been added as an attachment.
Comments and Suggestions for Authors:
1.After the title of the article, next to the last name of each author there should be only “1”, and not “2” and “3”, since the authors are from the same institution
Response:
The authors have made corrections in the manuscript
- Incomplete full abbreviation name.
Line 49: is easily weldable in the inert gas (TIG) and . It should be is easily weldable in the
tungsten inert gas (TIG) and .
Response:
The authors have made corrections in the manuscript
- When a term is first mentioned in the text of a manuscript, its full name should be used, not an abbreviation.
Line 56: tested 2050 aluminum alloy after the FSW process and . It should be: tested 2050
aluminum alloy after the friction stir welding (FSW) process and ,
Response
The authors have made corrections in the manuscript
- The aim of the study is presented in the Introduction section (lines 98 100) and repeated in Materials
and Methods section (lines 104 106). This should only be presented in the introduction.
Response
The authors have made corrections in the manuscript
A paragraph with a repeated research objective was deleted
- Filler metal chemical composition units not listed in Table 2.
Response
The authors have made corrections in the manuscript
- The main suggestion. The section them. The discussion of why the burnishing process increases the surface hardness of the welded joint, as well as a reasoned explanation of the reasons for this, significantly increased the scientific value of this study.
Response
An explanation of the change in hardness after the burnishing process is presented in the manuscript.
The basic features of metals and their alloys include surface hardness and microhardness, which are directly related to the properties of the surface layer, such as: resistance to frictional wear, fatigue strength, residual stresses and plasticity. The surface hardness depends on the type of material and the strengthening, thermal or thermo-chemical treatment carried out. The change in hardness and microhardness in the burnishing process is influenced by the burnishing tool, which, under the action of force, causes local elastic and plastic deformations in the contact zone with the workpiece. The phenomenon of strengthening mainly occurs in the surface layer. The grains are fragmented, flattened and elongated in the direction of the greatest deformations, creating a crushed texture. The ability of a material to be strain hardened depends largely on its structure and the value of burnishing parameters. The burnishing process is limited due to the permissible degree of plastic deformation and exceeding it results in technological cracking and flaking of the processed surface.
- The vast majority of cited references are older than five to ten years. This reduces the relevance of the work. It is advisable to include more references in publications less than five years old.
Response:
The authors have made corrections in the manuscript. The authors added 4 publications:
- Kluz R., Bucior M. , Dzierwa A., Ochał K., Antosz K., Bochnowski W. Effect of Diamond Burnishing on the Properties of FSW Joints of ENAW-2024 Aluminium Alloys. Materials 2023 , 13, 1305.
- Teimouria R.,Grabowski M.,Bogucki R., Ślusarczyka Ł., Skoczypiec S. Modeling of strengthening mechanisms of surface layers in burnishing process. Materials & Design 223 (2022) 111114.
- Felhȍ C., Varga G. 2D FEM Investigation of Residual Stress in Diamond Burnishing. Manuf. Mater. Process. 2022, 6, 123.
- Jerez-Mesa, R.; Fargas, G.; Roa, J.J.; Llumà, J.; Travieso-Rodriguez, J.A. Superficial Effects of Ball Burnishing on TRIP Steel AISI 301LN Sheets. Metals 2021, 11, 82.
- Typos:
Superfluous word: line 46 rods, rods and drawn tubes.
Correct the indices in the surface roughness parameters, hardness parameters, and also in the formulas.
For example: Al-Mg2Si Al-Mg2Si; Mg2Si Mg2Si; Su Su; Rq Rq; Rt Rt; Rz Rz; Vt
Vt and so on. Check all text carefully and make corrections.
Response
The authors have made corrections in the manuscript

Reviewer 2 Report
Comments and Suggestions for Authors
1. The authors must include the micro-structure of the weldment
2. The fracture toughness of the weldment must be evaluated
3. The fracture surface morphologies of the tensile and toughness tests must be presented and discussed
4. The X-ray stress measurements before and after burnishing must be made and discussed.
Comments on the Quality of English Language1. The authors must include the micro-structure of the weldment
2. The fracture toughness of the weldment must be evaluated
3. The fracture surface morphologies of the tensile and toughness tests must be presented and discussed
4. The X-ray stress measurements before and after burnishing must be made and discussed.
Author Response
Response to Reviewer Comments
Dear Reviewer,
Thank you very much for taking the time to review this manuscript.
Comments and Suggestions for Authors:
- The authors must include the micro-structure of the weldment
- The fracture toughness of the weldment must be evaluated
- The fracture surface morphologies of the tensile and toughness tests must be presented and discussed
- The X-ray stress measurements before and after burnishing must be made and discussed.
Response
Thank you very much for the valuable tips that we received in the review for publication. The literature review showed that issues related to burnishing of welded joints of EN AW-6060 aluminum alloys are not available in journal databases. Therefore, in the publication "The influence of the Burnishing Process on the Change of Surface Hardness, Selected Surface Roughness Parameters and the Material Ratio of the Welded Joint of Aluminum Tubes" the authors presented research that shows an improvement in the quality of the machined surface while strengthening the surface layer of the welded joint and native material. The authors fully agree with the reviewer's comments, which indicates that additional test results should be performed and presented to fully determine the impact of burnishing treatment on the joint and the native material. However, at the current stage of research, its impact on selected parameters of surface roughness, material fraction and surface hardness was determined. In the future, research is planned on the optimization of burnishing parameters, taking into account the quality of the machined surface and the strengthening of the surface layer, using the SRMD rolling burnisher from Yamato and then the hydrostatic burnisher from ECOROLL. Next, the research team plans to perform the tests indicated by the reviewer. In addition, corrosion resistance and contact fatigue tests will also be performed.
Reviewer 3 Report
Comments and Suggestions for Authors
The influence of the Burnishing Process on the Change of Surface Hardness, Selected Surface Roughness Parameters and the Material Ratio of the Welded Joint of Aluminum Tubes.
Very good manuscript.
Authors need to specify which type or article they are aiming to work here.
Rows 38 and 40, can you work on the subscripts?
Please consider taking out Figure 1. We all know what is a lathe, the figure 2 makes sense since you have the sample mounted in the lathe and that is the way you are going to work with the item...
Can you merge or couple Figures 3 and 4 in one single image? probably side by side...
Figure 5b, what is the scale you used here?
In your conclusions, you mentioned that further research must be done. What is the value added of this work then? compared to other published research on aluminum alloys...
I highly recommend the authors to search and identify more recent references, please use or work on those which are > 2021.
Comments on the Quality of English Language
English grammar needs minor work. Parragraphs reading flow could be improved with a review on grammar.
Author Response
Response to Reviewer Comments
Dear Reviever,
Thank you very much for taking the time to review this manuscript. Thank You for all the comments that will certainly affect the quality of our article, the corrections have been made in the article .
The publication with the comments taken into account has been added as an attachment.
Comments and Suggestions for Authors
- Authors need to specify which type or article they are aiming to work here.
Response:
The authors wrote a scientific article based on their research
- Rows 38 and 40, can you work on the subscripts?
Response:
The authors have made corrections in the manuscript
The publication has changed subscripts for marking materials, cutting parameters, surface roughness parameters, material ratio and index of relative surface strengthening.
- Please consider taking out Figure 1. We all know what is a lathe, the figure 2 makes sense since you have the sample mounted in the lathe and that is the way you are going to work with the item...
Response:
Of course, the authors agree with the opinion that lathes are commonly known machining machines. There are many manufacturers available on the market who offer many types . The publication presents a view of the cutting machine in order to provide full information about the research methodology, because currently CNC machine tools are often used in industry and scientific institutes. In this case, the process was carried out on an older generation conventional lathe.
- Can you merge or couple Figures 3 and 4 in one single image? probably side by side...
Response:
The authors have made corrections in the manuscript
- Figure 5b, what is the scale you used here?
Response:
In the publication, the authors presented the view of the impression from the hardness measurement carried out on the Qness 250M hardness tester with the HV10 load. The imprint is measured in [μm] and its diagonals are: diagonal 1 = 408.20 µm, diagonal 2 = 396.51 µm.
- In your conclusions, you mentioned that further research must be done. What is the value added of this work then? compared to other published research on aluminum alloys...
Response:
A review of the literature revealed few publications related to the burnishing process of welded joints of aluminum alloys. Moreover, soft materials pose problems during the burnishing process. In the publication, the authors presented research results that demonstrate an improvement in the quality of the machined surface while strengthening the surface layer of the welded joint and the native material.
The added value of the publication is that after welding, the EN AW-6060 aluminum alloy can be subjected to a burnishing process. The tests used burnishing parameters selected on the basis of experience and knowledge of the burnishing process with a Yamato SRMD burnishing tool. Further research will include the optimization of burnishing parameters in terms of the surface roughness reduction rate and the degree of surface strengthening. After determining the potential surface properties, tests of functional properties are planned using optimal burnishing parameters. Additionally, tests are planned using a hydrostatic burnisher from ECOROLL
- I highly recommend the authors to search and identify more recent references, please use or work on those which are > 2021.
Response:
Response:
The authors have made corrections in the manuscript. The authors added 4 publications:
- Kluz R., Bucior M. , Dzierwa A., Ochał K., Antosz K., Bochnowski W. Effect of Diamond Burnishing on the Properties of FSW Joints of ENAW-2024 Aluminium Alloys. Materials 2023 , 13, 1305.
- Teimouria R.,Grabowski M.,Bogucki R., Ślusarczyka Ł., Skoczypiec S. Modeling of strengthening mechanisms of surface layers in burnishing process. Materials & Design 223 (2022) 111114.
- Felhȍ C., Varga G. 2D FEM Investigation of Residual Stress in Diamond Burnishing. Manuf. Mater. Process. 2022, 6, 123.
- Jerez-Mesa, R.; Fargas, G.; Roa, J.J.; Llumà, J.; Travieso-Rodriguez, J.A. Superficial Effects of Ball Burnishing on TRIP Steel AISI 301LN Sheets. Metals 2021, 11, 82
